



# Divergence of apparent and intrinsic snow albedo over a season at a sub-alpine site with implications for remote sensing

Edward H. Bair[1], Jeff Dozier[2], Charles Stern[3], Adam LeWinter[4], Karl Rittger[5,1], Alexandria Savagian[6], Timbo Stillinger[1], and Robert E. Davis[4]

[1]Earth Research Institute, University of California, Santa Barbara, CA USA 93106
[2]Bren School of Environmental Science & Management, University of California, Santa Barbara, CA USA 93106
[3]Lamont-Doherty Earth Observatory, Palisades, NY USA 10964
[4]Cold Regions Research and Engineering Laboratory, Hanover, NH USA 03755
[5]Institute of Arctic and Alpine Research, University of Colorado, Boulder, Boulder, CO 80309
[6]Bowdoin College, Brunswick, ME USA 04011

*Correspondence to*: Edward H. Bair (nbair@eri.ucsb.edu)

**Abstract**

Intrinsic albedo is the bihemispherical reflectance of a substance with a smooth surface. Conversely, the apparent albedo is the

bihemispherical reflectance of the same substance with a rough surface. For snow, the surface is often rough, and these two optical quantities have different uses: intrinsic albedo is used in scattering equations whereas apparent albedo should be used in energy balance models. Complementing numerous studies devoted to surface roughness and its effect on snow reflectance, this work analyzes a timeseries of intrinsic and apparent snow albedos over a season at a sub-alpine site using an automated terrestrial laser scanner to map the snow surface topography. An updated albedo model accounts for shade, and in situ albedo

measurements from a field spectrometer are compared to those from a spaceborne multispectral sensor. A spectral unmixing approach using a shade endmember (to address the common problem of unknown surface topography) produces grain size and impurity solutions; the modeled shade fraction is compared to the intrinsic and apparent albedo difference. As expected and consistent with other studies, the results show that intrinsic albedo is consistently greater than apparent albedo. Both albedos decrease rapidly as ablation hollows form during melt, combining effects of impurities on the surface and increasing roughness.

Intrinsic broadband albedos average 7% greater than apparent albedos, with the difference being about 6% in the near-infrared or 3-4% if the average (planar) topography is known and corrected. Field measurements of spectral surface reflectance confirm that multispectral sensors see the apparent albedo but lack the spectral resolution to distinguish between darkening from ablation hollows versus low concentrations of impurities. In contrast, measurements from the field spectrometer have sufficient resolution to discern darkening from the two sources. Based on these results, conclusions are: 1) impurity estimates from

multispectral sensors are only reliable for relatively dirty snow with high snow fraction; 2) a shade endmember must be used in spectral mixture models, even for in situ spectroscopic measurements; and 3) snow albedo models should produce apparent albedos by accounting for the shade fraction. The conclusion re-iterates that albedo is the most practical snow reflectance quantity for remote sensing.



## 1. Introduction

Snow albedo plays an important role in Earth's climate and hydrology. For example, a small (1.5% to 3.0%) decrease in snow albedo over the Northern Hemisphere is twice as effective as a doubling of $CO_2$ at raising global air temperature (Hansen and Nazarenko, 2004). Likewise, during the COVID-19 lockdowns, a cleaner snowpack, presumably from a reduction in anthropogenic emissions, prevented 6.6 km$^3$ of snow/ice from melting in the Indus River Basin (Bair et al., 2021a), more water than is stored in the largest reservoir in California. Yet, snow albedo is difficult to measure (Bair et al., 2018), especially in the

mountains where lighting conditions vary dramatically. In order to understand Earth's climate and the effect humans have on it, an understanding of how snow surface topography affects snow albedo is imperative. The concepts of intrinsic and apparent albedos form the basis of this study. Intrinsic albedo is the bihemispherical reflectance (Nicodemus et al., 1977;Schaepman-Strub et al., 2006) of a substance with a smooth surface. Apparent albedo is the same, but for a rough surface. Here we use the term albedo to refer to a broadband albedo, covering the solar spectrum. Albedos covering a narrower spectral range are

denoted with additional descriptors such as near-infrared albedo. Since the snow surface is rarely smooth, distinction between apparent and intrinsic albedo is an important consideration that is often ignored. For example, MODIS measurements of snow albedo that comprise the National Solar Radiation Database have been found to be positively-biased because they fail to account for surface roughness (Gueymard et al., 2019). Both albedos should be studied, as apparent and intrinsic albedos have different uses. An apparent albedo should be used when modeling energy budgets (Bair et al., 2016), as it dictates how much

shortwave radiation is absorbed by the surface. Intrinsic albedos are needed to understand changes in snow properties that affect albedo, such as changes in grain size and darkening from light-absorbing particle like soot or dust (Clarke and Noone, 1985;Jones, 1913;Warren, 2019).

      Most snow albedo models follow approaches developed four decades ago, based on radiative transfer (Warren, 1982). These models provide intrinsic albedos with lighting conditions controlled by snow properties and illumination angles for a

smooth surface; they have been modified to consider grain shape (Libois et al., 2013), slopes (Picard et al., 2020), snow structure (Kaempfer et al., 2007), direct and indirect effects of light-absorbing particles (Skiles and Painter, 2019;Picard et al., 2020), and vertical heterogeneity (Zhou et al., 2003). Other efforts have focused on rapid calculation (Gardner and Sharp, 2010;Bair et al., 2019;Flanner et al., 2021) and inversion from remotely sensed imagery (Nolin, 2010;Painter et al., 2012a;Bair et al., 2021b). Weiser et al. (2016) present a correction for albedometers over snow where the underlying terrain is unknown,

based on modeled or measured irradiance from nearby well-leveled radiometers, but not accounting for surface roughness. A shade endmember has been introduced to account for lighting differences across surfaces (Adams et al., 1986), thereby enabling the use of an apparent albedo for quantitative spectroscopy. These shade endmembers have proved successful when applied to snow cover mapping (Painter et al., 2003;Bair et al., 2021b;Rosenthal and Dozier, 1996;Nolin et al., 1993). Yet, the widely used albedo models cited above do not account varying illumination within the field-of-view, meaning their results can

be positively biased.



Features that affect snow roughness include suncups (ablation hollows), penitentes, and sastrugi. Because of their topographic variation in solar exposure, all of these roughness features can significantly affect albedo. Matthes (1934) described "suncups" to have a "a honeycombed appearance, the surface being pitted with deep cell-like hollows..." However, Rhodes et al. (1987) use the term "ablation hollows" to describe these features as they are not always caused by solar radiation. Instead Rhodes et al. (1987) find that the presence of impurities on the snow surface governs the formation of ablation hollows, growing in direct sunlight for relatively clean snow and decaying in dirty snow (Lliboutry, 1964). This hypothesis was confirmed with a field experiment where an ash-covered snowfield on Mount Olympus from the Mount Saint Helen's eruption was cleared. After two weeks, the ash-free area had developed larger ablation hollows than the rest of the ash-covered snowfield (Rhodes et al., 1987). Observations of "penitentes" go back to Darwin (1845, ch. XV). Penitentes are columns of snow that point at the sun and are thought to be sublimation features (Betterton, 2001). Penitentes can be much larger than ablation hollows, with measured heights over 2 m (Lhermitte et al., 2014). "Sastrugi" are smaller-scale snow roughness features, formed by wind erosion and pointing in the direction of the prevailing winds (Seligman, 1936). Warren et al. (1998) report that sastrugi can reduce albedo by altering the angle of incidence for direct solar radiation and by trapping photons through multiple reflections.

Several studies have attempted to model the reflectance of roughness features with simple shapes (Leroux and Fily, 1998;Carroll, 1982;Zhuravleva and Kokhanovsky, 2011) with more recent studies employing ray tracing of three-dimensional surface models (Larue et al., 2020;Manninen et al., 2021). A few studies have focused on the surface roughness and the implications for remote sensing by incorporating multiple viewing geometries (Kuchiki et al., 2011;Nolin and Payne, 2007;Corbett and Su, 2015;Lyapustin et al., 2010). These approaches are well-suited toward expansive high-latitude snowpacks, but ill-suited towards dynamic mid-latitude snowpacks with mixed pixels where the snow cover can change substantially between satellite overpasses. The consensus in the literature is that roughness features can lower the snow albedo by up to 40%, but decreases of a few percent are more common. To our knowledge, none of these studies have tracked the snow surface topography throughout a snow season, nor have they examined the effects of snow surface topography on spectral mixture analysis.

## 2. Approach

### 2.1. Radiometric measurements

Albedos were measured (Figure 1) at CUES—Cold Regions Research and Engineering Laboratory and University of California, Santa Barbara Energy Site—on Mammoth Mountain, CA USA (Bair et al., 2015). To eliminate darkening from the ground, shadowing from vegetation, and effects from high zenith angles, only clear days with a deep, optically thick snowpack were examined. Radiometer measurements were taken at the satellite overpass time (Section 2.3). Irradiance was measured using a clear dome (285-2800 nm transmission) Eppley Precision Spectral Pyranometer (PSP) mounted 8 m above the ground to provide near 100% sky view. The ratio of diffuse to direct irradiance was computed using a Delta-T SPN1 Sunshine



Pyranometer, which has a slightly different response (400-2700 nm transmission) than the PSP. Because of the different response and biases (Habte et al., 2015;Wilcox and Myers, 2008) arising from issues such as thermal offsets (Haeffelin et al., 2001), only the diffuse ratio (used in the terrain correction described in Section 2.2) from the SPN1 was used. The irradiance measured by the PSP was split into direct and diffuse components using this ratio. Calculations using SMARTS v2.9.8 (Gueymard, 2019) provide an estimate of the spectral distribution of irradiance not subject to instrument error. We use the SMARTS simulations to adjust the measurements of the diffuse fraction from the SPN1 (400-2700 nm) to account for the diffuse fraction in the irradiance measurements from the PSPs with clear and near-infrared domes. The accuracy of an atmospheric radiation model depends on the accuracy of the estimates of the atmospheric properties, principally aerosols and water vapor. Errors in field radiometer measurements stem from calibration inaccuracies and siting of the instrument. The comparison between SMARTS and the measurements yields $R^2 \geq 0.99$ for both the PSP and the SPN1 (Figure 2 and Table 1), suggesting sufficient relative accuracy to make both instruments suitable for albedo measurement. However, there are only downlooking PSPs (no downlooking SPN1) so those instruments were used to measure the magnitude of irradiance and reflected solar radiation.

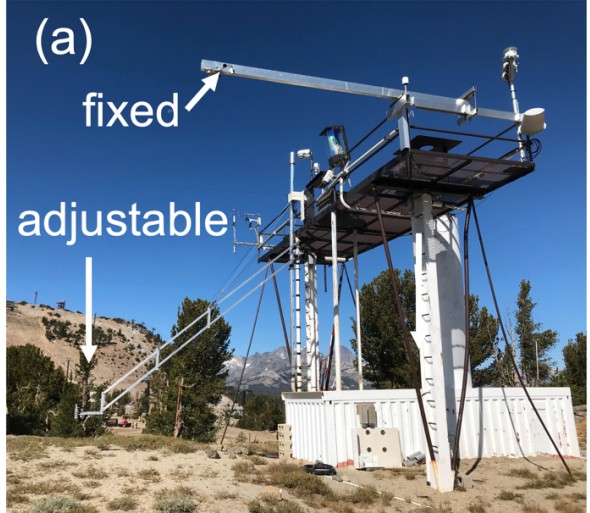
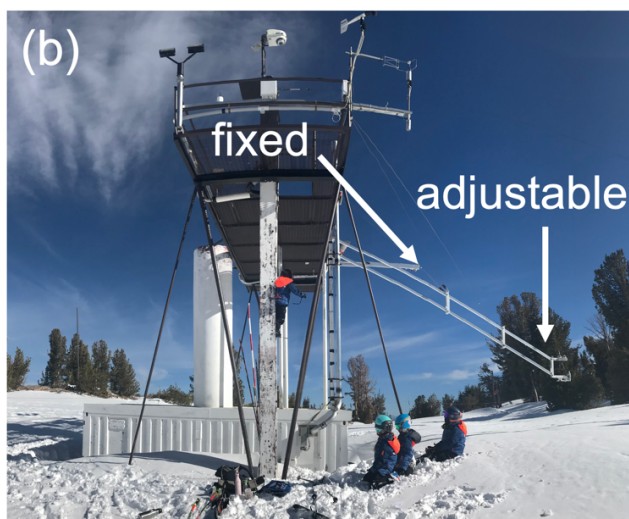

Figure 1:

*Fixed and adjustable albedo arms at the CRREL UCSB Energy Site (CUES) in the summer (a) and winter (b).*



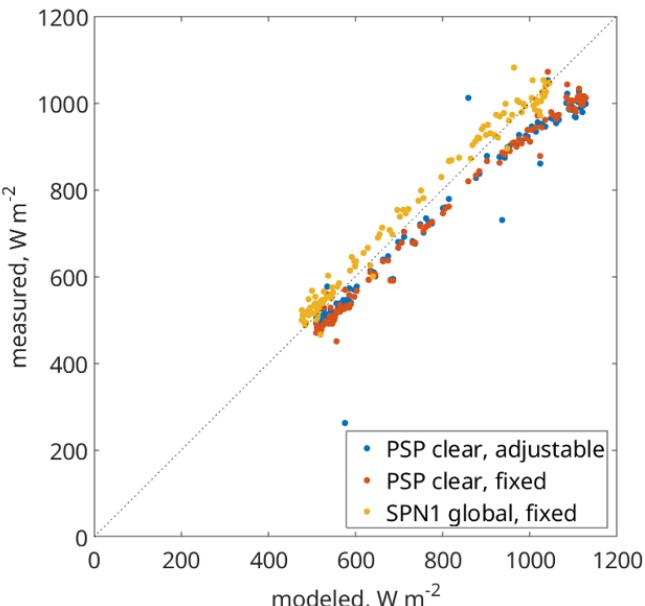

Figure 2:

*Measured vs. modeled irradiance at CUES for 3 broadband sensors: an Eppley Precision Spectral Pyranometer (PSP) mounted on an adjustable albedometer arm kept ~ 1 m above the snow surface (PSP clear, adjustable); a PSP mounted ~8 m above bare ground (PSP clear, fixed); and Delta-K SPN1 Sunshine Pyranometer also mounted ~8 m above bare ground (SPN1 global, fixed).*

| Name | RMS differences, W m$^{-2}$ | Difference, W m$^{-2}$ | $R^2$ |
|---|---|---|---|
| PSP clear, adjustable | 76 | -56 | 0.956 |
| PSP clear, fixed | 64 | -55 | 0.988 |
| SPN1 global, fixed | 35 | 22 | 0.984 |


Table 1:

*Radiometer measurement differences shown in Figure 2.*

Reflected radiation was measured with a downlooking PSP mounted on a computer-controlled and self-leveling arm. The downlooking PSP was kept ~1 m above the snow surface to prevent non-snow objects from being seen. To illustrate the effect

of non-snow objects within the downlooking radiometers' fields of view, Figure 3 shows a comparison.



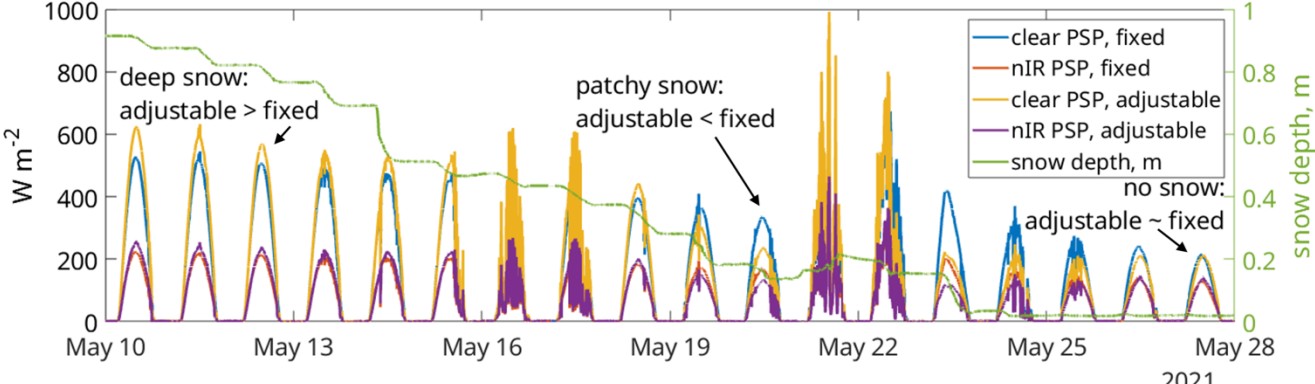

Figure 3:

*Reflected radiation, from downlooking radiometers, and snow depth measured at CUES.*

When the snowpack is deep and continuous spatially, the downlooking radiometers on the adjustable boom have greater
values than those on the fixed arm (Figure 3, 2021 May 10-17). This condition occurs because darker non-snow objects are
within the radiometers' fields-of-view on the fixed arm. Contrast this to the snow-free condition at the end of May where
reflected radiation is the same for the radiometers on both the fixed arm and adjustable arm. In patchy snow, the opposite
occurs; on 2021 May 19-20, the radiation measured by the nIR PSP on the fixed arm exceeds that of the clear PSP. This
condition occurs because the radiometers on the fixed boom view additional emerging vegetation with a higher nIR albedo
than snow. Thus, to prevent non-snow objects from contaminating the snow albedo measurements, only the downlooking
radiometers on the adjustable arm were used to measure reflected radiation.

Although a radiometer views a hemisphere, the downlooking field of view is restricted to about ~150º due to
manufacturing constraints (Wu et al., 2018). Sailor et al. (2006) showed that the size of a radiometer's field-of-view that
accounts for 95% of the reflected radiation is ~8.7$h$, where $h$ is the height of the radiometer above the surface. The
radiometer's height above the snow surface of $h$~1.0 m translates to a footprint diameter $d$ of 8.7 m. In comparison, the
downlooking radiometers on the fixed arm 8 m above bare ground would see a footprint larger than 40 m over snow with 1 m
depth. To our knowledge, CUES is the only site where snow albedo is measured using an adjustable albedo arm. Given such
a large footprint, an examination of published images of tower arms at other sites where snow albedo is measured (Lejeune et
al., 2019;Landry et al., 2014;Lhermitte et al., 2014;Elder et al., 2009) shows non-snow objects within the downlooking
radiometer's field-of-view at every site.

## 2.2.  Surface topography and corrections

A Riegl VZ-400 laser scanner automatically scanned the snow surface every hour during the 2021 water year. Point clouds
were converted to surfaces as follows. Noise was removed using a filter (Rusu et al., 2008) and additional days with blowing
snow were manually removed because the moving particles obscure the snow surface (Bair et al., 2012). The adjustable





albedometer arm was removed from the point clouds using a morphological filter (Pingel et al., 2013). Point clouds were converted to surfaces with 1 cm spatial resolution using bilinear interpolation. A radial mask was applied to the surface to simulate the footprint seen by the downlooking PSP. Slope and aspect were computed for a plane fit to the surface. For a measure of surface roughness, the root mean squared value of the distribution of slopes in the field-of-view was used. The surface roughness combines with the local illumination angle to affect the apparent snow albedo.

Four broadband albedos were computed. An uncorrected apparent albedo is computed as

$$\alpha_{uncorrected} = \frac{D_\uparrow}{I_\downarrow},$$ (1)

where $D_\uparrow$ is the reflected radiation measured by the downlooking PSP and $I_\downarrow$ is the irradiance measured by the uplooking PSP. An albedo with a plane fit to the surface built from the point cloud is computed as

$$\alpha_{planar} = \frac{D_\uparrow}{cB_\downarrow + D_\downarrow},$$ (2)

where $c = \cos\theta_S / \cos\theta_0$ is a correction factor of a sloped to a level surface. $\theta_S$ is the illumination angle for the plane, $\theta_0$ is the solar zenith angle for a level surface, $B_\downarrow$ is the direct irradiance, and $D_\downarrow$ is the diffuse irradiance. This planar correction has

been applied in previous work (Painter et al., 2012b;Bair et al., 2018). Because the ratio $c$ is in the denominator of Eq. (2), $\alpha_{planar} > \alpha_{uncorrected}$ when $\cos\theta_S < \cos\theta_0$, equal when the angles are equal, less otherwise.

An albedo with a spatial correction to account for the rough surface is computed by considering the effect of a generic ablation hollow and then averaging those effects over the downlooking radiometer's field-of-view, i.e., a circle with 8.7 m diameter. A point on the rough surface has slope $S$ and aspect $A$, and $\phi_0$ is the solar azimuth. The cosine of the illumination

angle at the point is

$$\cos\theta_S = \max[0, \cos\theta_0 \cos S + \sin\theta_0 \sin S \cos(\phi_0 - A)].$$ (3)

The use of the max function sets the value of $\cos\theta_S$ to zero on self-shaded slopes, when otherwise the cosine would be negative. In addition to the slope affecting the magnitude of the irradiance, local horizons formed by adjoining ablation hollows affect the illumination in two ways: (1) a neighboring high point might shade a slope that would otherwise be illuminated; (2) the set of horizons in all directions partly obstructs the overlying hemisphere. We define the view factor $V_\Omega$ as the fraction of

the hemisphere that is open to the sky; a completely unobstructed surface has a view factor $V_\Omega = 1$. Dozier (2021) describes methods to rapidly compute the horizons and the view factor.

Considering the albedo of a rough snow surface involves multiple reflections. Over a range of wavelengths, the spectral distribution changes with each reflection. Therefore, the initial approach the model this effect uses monochromatic radiation, with $\rho$ to indicate a spectral albedo, omitting a wavelength identifier unless necessary. Following that analysis, measured

wavelength-integrated albedos are used. Setting $F_{dif}$ as the fraction of the spectral irradiance that is diffuse and setting $I = 1$ as the value of the initial irradiance, the "spatial" spectral radiation that initially escapes into the overlying hemisphere without being re-reflected is:



$$I_{esc}^{(0)} = V_\Omega \left[ \frac{\cos\theta_S}{\cos\theta_0} \left(1 - F_{dif}\right)\rho_{intrinsic}^{(direct)} + F_{dif}\rho_{intrinsic}^{(diffuse)} \right] \tag{4}$$

$$\text{directly reflected} \qquad\qquad \text{diffusely reflected}$$

with $\rho_{intrinsic}$ as the intrinsic spectral albedo on a level smooth surface unaffected by topography; the superscripts designate the albedo to direct vs. diffuse irradiance. The direct and diffuse spectral albedos of snow differ slightly (Wiscombe and

Warren, 1980), the major difference in the broadband values lies in the different spectral distributions of the direct and diffuse irradiance. Generally, $\alpha_{intrinsic}^{(diffuse)}$ will be larger because the diffuse irradiance more heavily concentrates in the wavelengths where snow is brightest.

Not all the initially reflected radiation escapes into the overlying hemisphere. Instead, some of it re-reflects and eventually escapes or is trapped (Warren et al., 1998) by the roughness. The re-reflected radiation that does not escape is subject to

possible internal reflection, its initial value being:

$$I_{internal}^{(0)} = I_{esc}^{(0)} \left(\frac{1 - V_\Omega}{V_\Omega}\right). \tag{5}$$

To account for multiple reflections, at each reflection the value of the incident radiation is multiplied by the fraction $(1 - V_\Omega)$ that accounts for the reflection remaining in the ablation hollow, the fraction $V_\Omega$ that escapes, and the spectral albedo. The albedo of the re-reflected radiation, $\alpha_{intrinsic}^{(RR)}$, is biased toward the wavelengths where snow is brightest. An orders-of-scattering approach to the multiple reflections lets some reflected radiation escape at each iteration $n$ and some remains

available for re-reflection:

$$\text{escaped } I_{esc}^{(n)} = I_{internal}^{(n-1)}\rho_{intrinsic}^{(diffuse)}V_\Omega$$

$$\text{remaining } I_{internal}^{(n)} = I_{internal}^{(n-1)}\rho_{intrinsic}^{(diffuse)}(1 - V_\Omega). \tag{6}$$

This series converges in a half dozen iterations because $I_{internal}^{(n)}$ declines in proportion to $(1 - V_\Omega)^n$. The spatial spectral albedo $\rho_{spatial} = \sum I_{esc}/I$ (where $I = 1$).

To compare modeled and measured albedo integrated over a range of wavelengths—for example the broadband and near-infrared albedos described in Section 2.1—$\rho_{intrinsic}$ cannot simply be replaced with $\alpha_{intrinsic}$ in Eqs. (4) through (6), because

wavelength-integrated albedo depends on the convolution of the spectral albedo with spectral distribution of the irradiance. Including the spectral identifier $\lambda$, the wavelength integrated albedo is:

$$\alpha = \frac{\int_{\lambda_1}^{\lambda_2} \rho(\lambda)I(\lambda)d\lambda}{\int_{\lambda_1}^{\lambda_2} I(\lambda)d\lambda} \tag{7}$$

where $\rho(\lambda)$ varies with wavelength, so $\alpha I$ has a different spectral distribution than $I$ itself. That distribution is weighted toward the wavelengths where $\rho(\lambda)$ is larger, so each reflection causes $\alpha$ to increase even though $\rho(\lambda)$ does not change. To address this problem, an empirically-derived function is used to estimate intrinsic broadband and near-infrared albedos at step $n$. In

Eqs. (4) through (6), $\rho_{intrinsic}$ is replaced with $\alpha_{intrinsic}^{(n)} = f(\alpha_{intrinsic}^{(0)}, n)$. The modeled average $\overline{\alpha_{spatial}}$ is equivalent to the





measured $\alpha_{uncorrected}$, so comparing the model to the measurement enables solving for the intrinsic wavelength-integrated snow albedo $\alpha^{(0)}_{intrinsic} = \alpha_{intrinsic}$.

To create $f(\alpha^{(0)}_{intrinsic}, n)$, solar irradiance spectra were generated using SMARTS (Gueymard, 2019) over observed solar zenith angles, 23° to 63°. Spectral snow albedo was simulated (Warren, 1982) over the range of zenith angles, snow grain
effective radii from 50 μm to 1000 μm, and mass concentrations of San Juan dust (Skiles et al., 2017) from $10^{-8}$ to $10^{-3}$ (i.e., 10 ng/g to 1 g/kg), assuming an effective dust radius of 3 μm. The simulation thus covered spectral albedo ranges of clean to dirty snow with fine to coarse grains. The SMARTS calculations also enabled transformation of the diffuse fraction measured by the SPN1 to the wavelength ranges of the broadband and near-infrared radiometers. Eq. (4), without the $V_\Omega$ term, was applied and spectral albedos were multiplied by the spectral irradiance. Defining $I$ as spectral radiation and $E$ as wavelength-
integrated radiation, initial values are:

$$I^{(0)}_{reflected}(\lambda) = I_\downarrow(\lambda)\{\rho_{direct}(\lambda)[1 - F_{dif}(\lambda)] + \rho_{diffuse}(\lambda)F_{dif}(\lambda)\}$$

$$E^{(0)}_{reflected} = \int_{\lambda_1}^{\lambda_2} I^{(0)}_{reflected}(\lambda)\, d\lambda \tag{8}$$

$$\alpha^{(0)} = E^{(0)}_{reflected} / \int_{\lambda_1}^{\lambda_2} I_\downarrow(\lambda)\, d\lambda.$$

Then at each iteration, the value of $\alpha$ increases in the following way (Figure 4). Note that $V_\Omega$ is omitted from these iterations, because the interest lies in the change in wavelength-integrated albedo, not in the escaping radiation at each reflection. Moreover, all the radiation in the subsequent reflections is diffuse,

$$I^{(n)}_{reflected}(\lambda) = I^{(n-1)}_{reflected}(\lambda)\rho_{diffuse}(\lambda)$$

$$E^{(n)}_{reflected} = \int_{\lambda_1}^{\lambda_2} I^{(n)}_{reflected}(\lambda)\, d\lambda \tag{9}$$

$$\alpha^{(n)} = E^{(n)}_{reflected} / E^{(n-1)}_{reflected}.$$

The assumption of a Lambertian surface versus the use of directional quantities differs in the snow literature. In this study,
a Lambertian assumption is used, justified with the use of nadir looking instruments with measurements taken midday and with the lack of directional knowledge of the re-reflected radiation. Further, as surface roughness increases, so does backscattering (Manninen et al., 2021), thereby counteracting some of the forward scattering in snow. Finally, ablation hollows, the largest surface roughness features observed, have no preferred orientation, unlike sastrugi or penitentes. These factors reduce the importance of angular effects (Painter and Dozier, 2004;Warren et al., 1998). Further, a goal of this study is
to compare in situ with remotely sensed snow measurements. At the remote sensing scale, the average or sub-pixel scale snow surface topography is usually unknown, thus the directional factors cannot be accurately computed. Although the snow-free topography may be known, the snow surface above can differ markedly especially at fine (e.g., meter) scales.



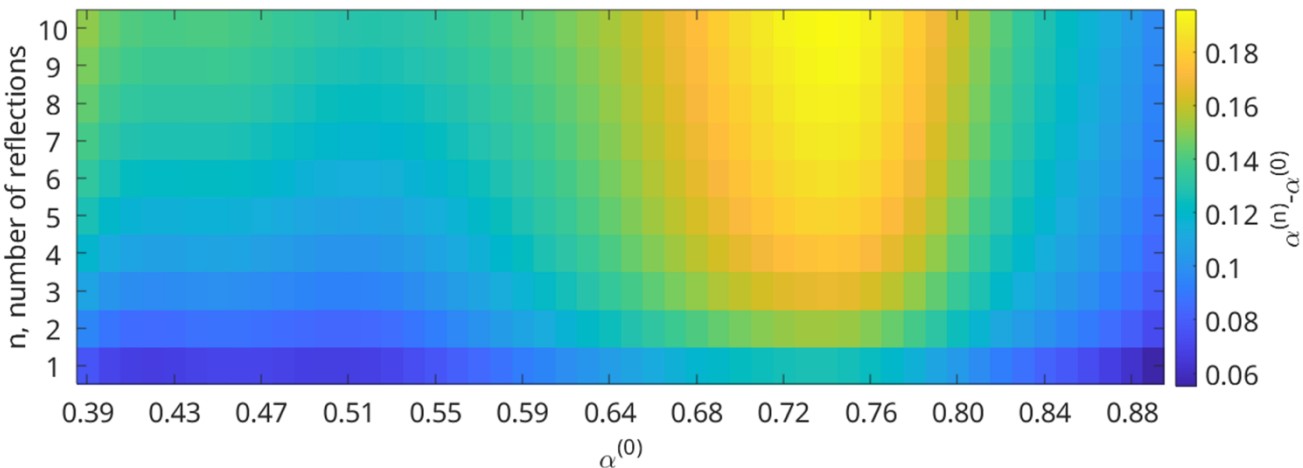

Figure 4:

*Increase in broadband albedo caused by internal reflections within an ablation hollow. The x-axis show the initial albedo from Eq. (8) and the y-axis show the number of reflections from 1 to 10. The intensity shows the resulting increase in albedo from Eq. (9).*

## 2.3. Remotely sensed measurements

Bottom of atmosphere (surface, Level 2A) reflectance estimates from the Sentinel-2A/B (S2) Multispectral Instrument were obtained. Nine bands (bands 2-7, 8a, and 11-12) were used with a spatial resolution of 20 m. To convert the narrow band surface reflectance estimates to broadband albedo, coefficients for snow-free and snow-covered surfaces, derived from radiative transfer simulations were used (Table 2 in Li et al., 2018). This surface reflectance product was processed using the Snow Property Inversion from Remote Sensing (SPIReS, Bair et al., 2021b) to obtain fractional snow-covered area and surface properties. Broadband albedo uncertainty from S2 (3.6%) was estimated based on maximum differences between acquisitions for a bare-ground target pixel, consisting of no trees, bare soil, and small shrubs. This uncertainty is close to a validation effort of S2 over dark and bright soils that showed band-wise errors up to 4.0% (Gascon et al., 2017).

The target pixel on Mammoth Mountain for comparison to the snow measured at CUES was selected because it is near CUES (2.2 km away), is at a similar elevation (CUES at 2916 m vs. target at 3041 m), has a slope of zero across the 20 m pixel, and was nearly 100% snow-covered for 6 months, from mid-November through mid-May. It would have been preferrable to select a pixel immediately adjacent to CUES, but none met those criteria. Thus, it is assumed that snow conditions and thus albedo were similar at the two sites, at least within the uncertainty of the remotely sensed and in situ broadband measurements. The mean local solar time for overpass from Sentinel-2 is 10:30, leading to times at CUES of 18:39 to 18:47 UTC. Thus, the corresponding in situ albedo measurements described in Section 2.2 were taken within that window of time.





### 2.4. Shade endmember simulations

Intrinsic snow albedo was modeled using a two-stream radiative transfer approximation coupled with Mie scattering as described in Section 2.2. Of note is that dust is assumed to be the predominant pollutant, based on chemical analyses from CUES (Sterle et al., 2013). Other endmembers used were an empirical snow-free background (for the remotely sensed solutions), and an ideal shade endmember with an albedo of zero across all bands (Adams et al., 1986).

### 2.5. In situ spectroscopy

A Spectra Vista HR-1024i was used with a 99% Spectralon panel for irradiance measurement. The lens used has a 4º field of view and was held about 1.5 m above the snow surface, leading to a footprint of about 5 cm. Measurements were made on days with clear skies and the spectrometer was held plumb rather than slope parallel. Noise was smoothed using an 11-point sliding window fit with a local regression using a 1st degree polynomial.

### 3.   Results and discussion

An example of ablation hollows mapped by the laser scanner is shown in Figure 5ab.

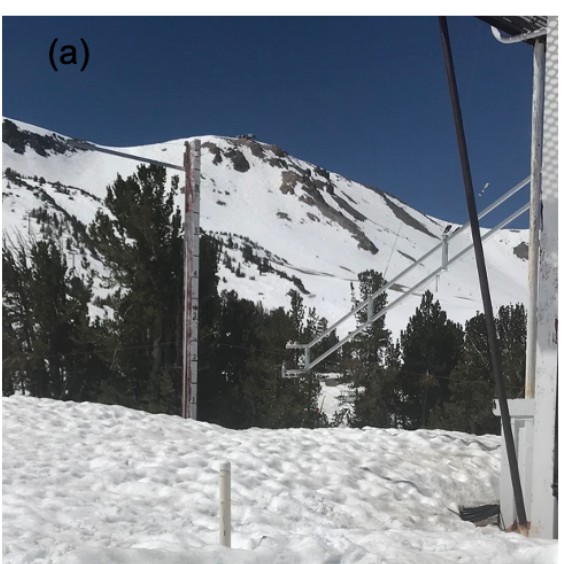
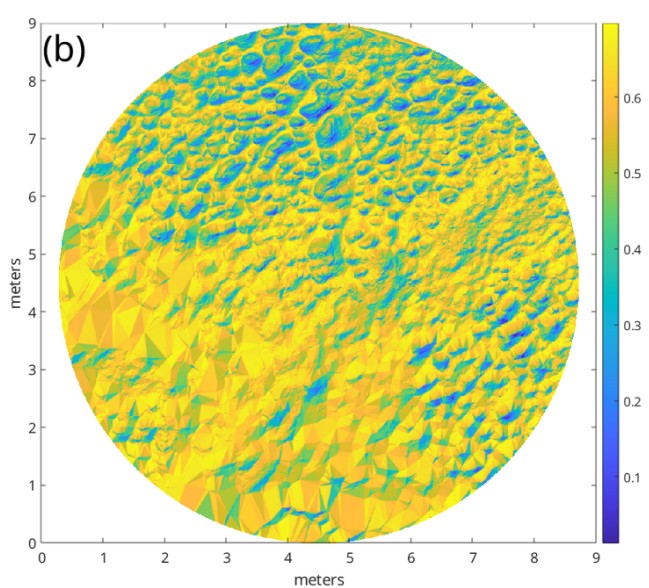

Figure 5:

*Snow with ablation hollows 12 May 2021 10:45:00 PST (a) Corresponding apparent albedo seen by the radiometer (b) The uncorrected albedo is 0.54 (mean of what is shown). The albedo with a planar correction is 0.55 and the intrinsic albedo based on the spatial analysis is 0.65.*

In situ albedos from CUES from water year 2021 are shown in Figure 5: uncorrected $\alpha_{uncorrected}$, planar corrected $\alpha_{planar}$, and intrinsic $\alpha_{intrinsic}$ based on the spatial calculations.





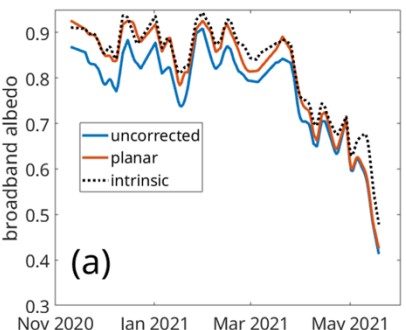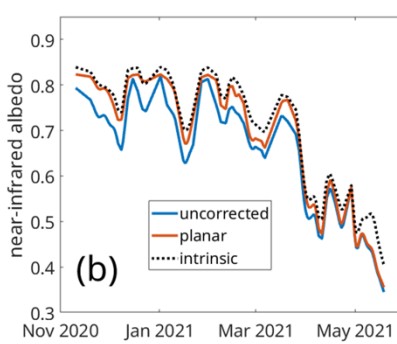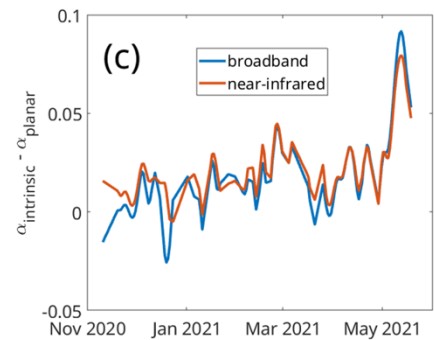

*Figure 6. In situ albedos on Mammoth Mountain in water year 2021. Shown are the uncorrected, planar corrected, and intrinsic albedos for broadband (a) and near-infrared (b) wavelengths. Planar correction involved fitting a plane to the snow surface*
*and using the solar illumination angle on that plane compared to that on a flat surface. Intrinsic albedos are derived from analyzing the view factors and illumination angles on the rough surface and using Equations (4) through (9) to solve for $\alpha_{intrinsic}$. The difference between the intrinsic and planar albedos in shown in (c).*

In situ and remotely sensed albedos on Mammoth Mountain from water year 2021 are shown in Figure 7. An unadjusted (i.e.,

not adjusted for shade or trees) fractional snow-covered area (fsca), estimated with SPIReS (Bair et al., 2021b), from a nearby

target pixel are also shown. The high fsca confirms that mixed (snow and non-snow) pixels effects are minimal. An estimate

of the broadband pixel albedo measured by Sentinel 2A/B (S2) is also shown, as described in Section 2.3. Finally, the surface

roughness (in degrees, divided by 30 for scale) is plotted, also described in Section 2.2.

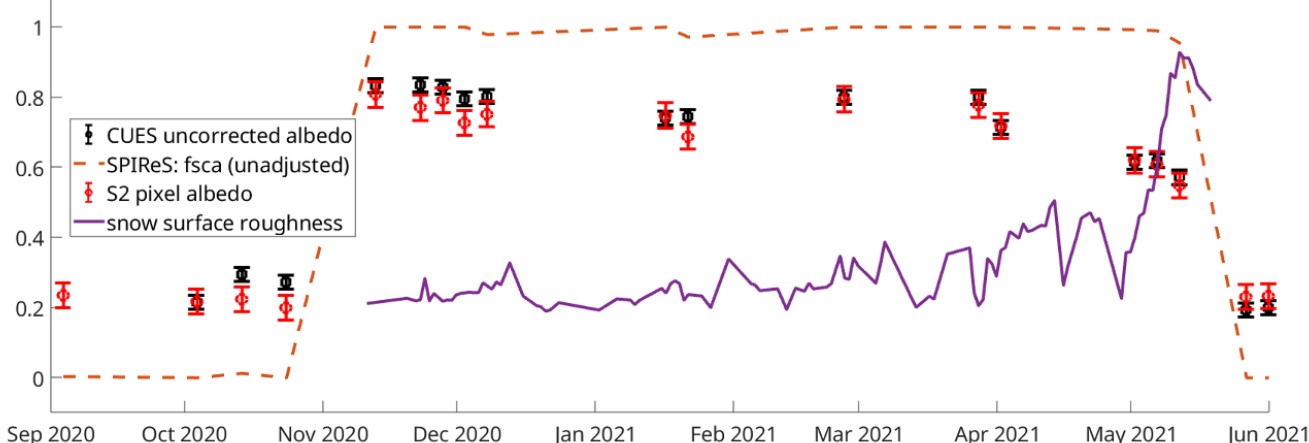

Figure 7:

*In situ and remotely sensed snow on Mammoth Mountain, water year 2021. Shown are uncorrected albedos measured at CUES, with the error bars (2.0 %) representing based on stated values from the manufacturer. The unadjusted (i.e., not adjusted for shade or trees) fractional snow-covered area (fsca) from the Snow Property Inversion from Remote Sensing (SPIReS) model is shown. An estimate of the broadband pixel albedo measured by Sentinel 2A/B (S2) is shown. The error bar height (3.6 %) is the maximum difference in the bare ground (no snow) reflectance. Last, the surface roughness (in degrees,*
*divided by 30 for scale) is plotted, which is the root mean squared value of the slope of the snow surface.*

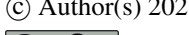



In Figure 6, the intrinsic albedo is usually greater than the uncorrected or planar corrected albedo, agreeing with previous work over more limited timespans (e.g., Lhermitte et al., 2014;Larue et al., 2020;Manninen et al., 2021). The largest planar corrections appear in winter, when the planar sloped surface facing away from the sun is receiving the lowest irradiance relative to a flat surface. The spatial corrections are more nuanced because they involve the solar geometry and the roughness of the surface. As the days get longer in the spring, the solar zenith angle is smaller, but the rougher surface causes more variability in the view factor and illumination on each slope.

Warren et al. (1998) posited two mechanisms for albedo reduction caused by surface roughness: reduction of effective illumination angle and photon trapping. The difference $\alpha_{uncorrected} - \alpha_{intrinsic}$ characterizes the combined contribution. In this study covering 110 days of the water year 2021 snow season, the differences amounted to –7% in the broadband albedo and –6% in the near-infrared. Larue et al. (2020) estimate a decrease of spectral albedo at 1000 nm of -2 to -3% for low SSA (i.e., large grain size) snow, but in the snow studied here with extensive ablation hollows, the magnitudes are greater. The difference $\left[I_{internal}^{(0)} - \left(\sum I_{esc} - I_{esc}^{(0)}\right)\right]$ characterizes photon trapping, which accounts for a mean of 3-4% of the lost broadband radiation and 5% of the loss in the near-infrared. This results follows from Warren et al. (1998), who state that intermediate snow albedos will be most impacted by photon trapping.

The intrinsic albedo is generally greater than the planar-corrected albedo, showing that the planar correction that has been performed in previous research (Bair et al., 2018;Painter et al., 2012b) accounts for surface slope but not for roughness. But the planar correction is useful as the difference between the planar corrected and the intrinsic albedo quantifies the impact of sub-slope surface roughness at this location. This difference implies that in areas where the average surface topography is accurately quantified (e.g., over 0.5-1.0 km pixels), a terrain-corrected (adjusted to level) surface reflectance can be used in a spectral mixture model in with a shade endmember to decrease uncertainty in impurity estimates. However, for sensors with finer resolution (e.g., ≤ 30 m), caution is advised with terrain corrections. If ground control points are not available, as in the case of many remote parts of the world, vertical errors in high-resolution elevation products approach the pixel size (Gottwald et al., 2017;Rodríguez et al., 2006;Shean et al., 2016). These errors are compounded when computing gradients (i.e., slope and aspect) needed for terrain corrections. These errors are especially noticeable for sharp features such as ridgelines. Thus, a shade endmember without any terrain correction may produce the most accurate results for these locations.

Narrow-to-broadband albedo conversions confirm that the apparent albedo is being seen from space. As surface roughness increases to its maximum during melt, albedo falls rapidly. This time period coincides with the time of year when snow becomes dirtiest on the surface, as the albedo is no longer being refreshed with new snowfall. Thus, the darkening effects of surface roughness occur simultaneously with the build-up of impurities (Betterton, 2001;Rhodes et al., 1987), which presents a challenge for remote sensing. However, because impurities only affect visible through near-infrared snow albedo, and snow grain size only affects albedo in the nIR/SWIR, while shadowing affects the entire broadband spectrum, an instrument with sufficient spectral resolution and accuracy should be able to discriminate between the causes of darkening.



To test this hypothesis, SPIReS was run on S2 imagery with dirty snow endmembers and with a clean snow assumption.
The resulting grain size and impurity concentration estimates were then used in the updated broadband snow albedo that now
accounts for shade. Because the pixel is close to fully-snow covered, this estimated albedo should be comparable to the narrow-
to-broadband conversions shown in Figure 7. The uncorrected albedo measured at CUES from Figure 7 is plotted along with
these two model runs (Figure 8). With overlapping error bars for each scene, the resulting albedos are indistinguishable within
measured error (Bair et al., 2021b). In the clean-snow run, the dust endmember is swapped for the shade endmember (Table
2). In situ spectroscopic measurements (also in Table 2) provide some validation, but also illustrate the wide spatial variability
of the snow surface just across the CUES study area.

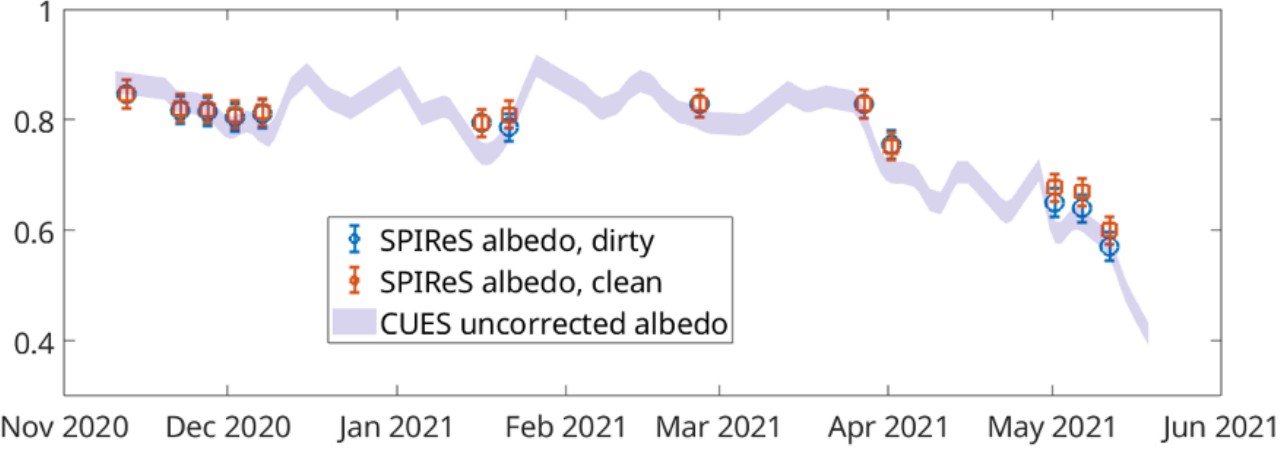

Figure 8:

*Broadband snow albedo solutions from SPIReS compared to the ucorrected albedo measured at CUES (same as in Figure 7).*
*In the first set of SPIReS solutions, dirty snow endmembers are used, while in the other the snow is assumed clean. Both sets*
*use a shade endmember. Error bars are $\pm2.5\%$ (Bair et al., 2021b).*

| Instrument | Dirty or Clean Snow Assumed? | Albedo | fsca | fshade | Grain Radius, µm | Dust, ppm |
|---|---|---|---|---|---|---|
| S2 | dirty | 0.55-0.60 | 0.96 | 0.00 | 766 | 122 |
| S2 | clean | 0.57-0.62 | 0.77 | 0.23 | 130 | 0 |
| SVC | dirty | 0.41-0.63 | 0.63-0.94 | 0.06-0.37 | 453-538 | 48-282 |

Table 2:

*Model solutions from SPIReS using measurements from Mammoth Mountain taken on 2021 May 11, the last 2 points with*
*error bars shown in Figure 8. The instruments are Sentinel 2B MSI (S2) and the Spectra Vista HR 1024i field spectrometer*
*(SVC). One of the SPIReS runs used a clean snow assumption to illustrate the difficulty in separating shade from dust*
*endmembers (with low concentrations) with a multispectral instrument. The fractional snow-covered area (fsca) and shade*
*(fshade) as well as the grain radius and dust concentration are unknowns that are solved for.*





Importantly, the spectroscopic measurements show that, when used in a model, there is a consistent ability to discriminate between darkening caused by impurities and by shade. For example, despite the high spatial variability, neither the shade endmember nor the dust concentration is zero in any of the solutions. An example of a dirty snow with a shaded solution is shown in Figure 9.

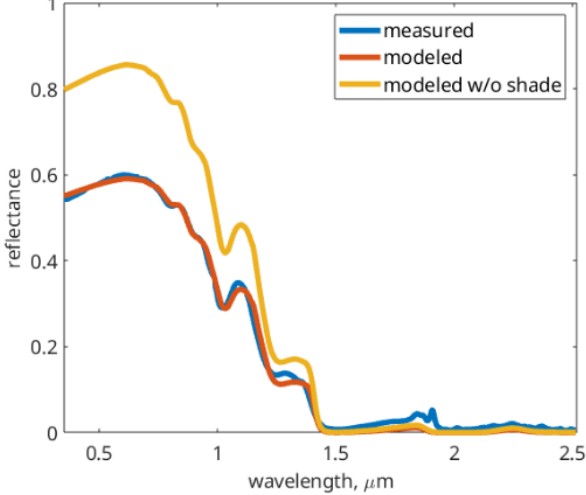

Figure 9:

*Example of measured and modeled reflectance from field spectroscopy measurements from 12 May 2021 (Table 2). The model estimates (with an RMSE = 0.006) are: fshade = 0.31; grain radius = 454 μm; dust concentration = 77 ppm; $\alpha_{apparent}$ = 0.45 (measured/modeled); $\alpha_{intrinsic}$ = 0.66 (modeled w/o shade).*

Because the snow surface is rarely flat or level, shade needs to be accounted for, even when using measurements taken from a
field spectrometer. Thus, shade needs to be included in snow albedo models, which often use look up tables for rapid processing. Figure 10 shows the results of radiative transfer simulations to illustrate the effect of shade on the difference between intrinsic and apparent albedo.



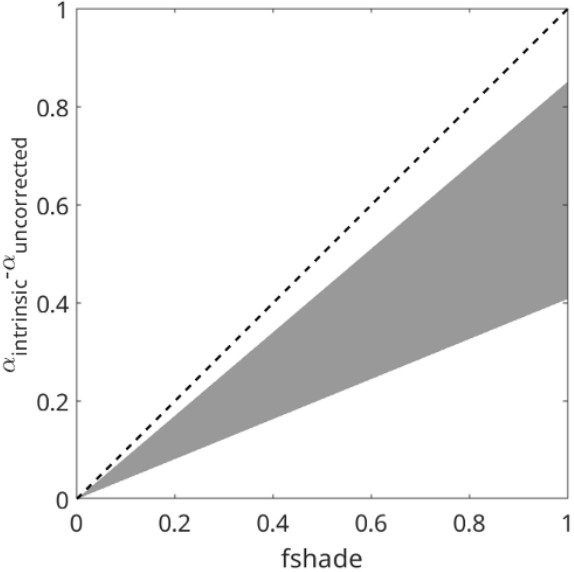

Figure 10:

*Difference between intrinsic and apparent albedo versus shade fraction. The gray area represents the range of radiative transfer solutions using different combinations of grain sizes, solar zenith angles, and impurities.*

There is a positive relationship, as fshade increases, the difference between intrinsic-apparent albedo increases, but the scatter also increases. A simple adjustment is not possible; instead the look up tables and albedo model presented in Bair et al. (2019) have been updated to include fshade. The new albedo model estimates an apparent albedo as

$$\alpha_{apparent} = f\left(r_g, \mu, Z, LAP_{name}, \delta, 1 - f_{shade}\right) \tag{1}$$

where $\alpha_{apparent}$ is the apparent albedo over three wavelength ranges (broadband, near-infrared, and visible), $r_g$ is the grain radius in µm, $\mu$ is the cosine of the solar zenith angle, $Z$ is the surface elevation in km, $LAP_{name}$ is the type of light-absorbing particles (dust or soot), and $\delta$ is the LAP concentration. Other properties such as an assumed mid-latitude winter atmosphere are unchanged from Bair et al. (2019).

## 4. Conclusion

A timeseries of intrinsic and apparent snow albedos over a season at a subalpine site were presented. In situ albedo measurements were compared to those from a spaceborne multispectral sensor. The multispectral measurements and those from a field spectrometer were used in a spectral mixture model. As expected and consistent with other studies, the results show that intrinsic albedo is consistently greater than apparent albedo. Both albedos decrease rapidly as ablation hollows form during melt, combining effects of build-up of impurities on the surface and increasing roughness.

There are several conclusions with implications for remote sensing, but also in situ measurement of snow albedo. For multispectral sensors, darkening effects from snow surface roughness are significant and can easily be confused with those





from impurities. In contrast, measurements from a field spectrometer have sufficient spectral resolution and accuracy to distinguish between the two effects. A spectral mixture model run on spectra obtained at a study site confirms significant darkening at the snow surface, simultaneously occurring from roughness and impurities, with wide variation spatially. In turn, 370   a spectral mixture model was used with Sentinel 2A/B multispectral imagery assuming a clean snowpack and a dirty snowpack. Both model runs were able to match measured snow albedo with plausible solutions, but the clean snow model used the shade endmember in place of the dust endmember.

    The -7% difference between apparent and intrinsic albedo is equivalent to the decrease in broadband albedo caused by 120 ppm dust for typical snow in spring. If the surface topography is known to the point where a plane can be fit, the difference 375   between the planar-corrected and the intrinsic albedo (mean of -3 to -4%) could be used instead, equivalent to darkening by around 50 ppm dust. Thus, to improve uncertainty in impurity estimates, a terrain correction used in conjunction with a shade endmember in a spectral mixture model can be used for moderate resolution sensors (e.g., 0.4 - 1 km), but caution is advised for terrain corrections at finer resolutions ($\leq$ 30 m) due to elevation model errors. Generally, impurity estimates from multispectral sensors are only distinguishable from surface roughness effects for relatively dirty snow. Likewise, for a 380   multispectral sensor, mixed pixels can be spectrally inseparable from pixels containing only dirty snow. Thus, only pixels with high snow fraction should be used for impurity estimates from a multispectral sensor (Bair et al., 2021b;Painter et al., 2012a). These conclusions were also reached by Warren (2013), but for black carbon on the snow surface in the Arctic.

    This study emphasizes the difficulties in modeling lighting conditions on the snow surface. Because of these difficulties, a recommendation is to always use a shade endmember in unmixing models, even for in situ spectroscopic measurements. 385   Likewise, snow albedo models should produce apparent albedos by accounting for the shade fraction. To this end, lookup tables and code have been revised to account for shade. The apparent albedo produced should be used in energy balance models where intrinsic albedos have been previously used.

    In this study, albedos were used rather than directional reflectance quantities. The justifications are: the use of nadir looking instruments with measurements taken midday; that as surface roughness increases, so does backscattering, thereby 390   counteracting the forward scattering in snow; and that ablation hollows, the largest surface roughness features observed, have no preferred orientation, unlike sastrugi or penitentes. These factors reduce the importance of angular effects. But perhaps the most compelling justification is that for snow, the average or sub-pixel scale snow surface topography is usually unknown, so the directional factors cannot be accurately computed.

    Future work could focus on testing these findings in other snow climates with different surface roughness features. The 395   ablation hollows studied are larger than sastrugi but smaller than penitentes, so the darkening effects are likely intermediate also. The findings about discrimination between darkening from surface roughness and impurities as well as detection limits for impurities from multispectral sensors requires further testing. For example, results from dirtier snowpacks should be examined, although the size of the ablation hollows will be reduced (Rhodes et al., 1987;Lliboutry, 1964). These findings highlight the need for hyperspectral measurements of snow from aerial and spaceborne sensors. The NASA Earth Observing- 400   1 Hyperion was promising in this regard, but lack of coverage, repeat passes, or a surface reflectance product limited utility.



The upcoming NASA Surface Biology and Geology (SBG) and ESA Copernicus Hyperspectral Imaging Mission for the Environment (CHIME) spaceborne spectrometers may offer chances to test these findings using spectroscopic measurements from space.

**Code availability**

All of the code used is available on GitHub at the first author's repository: https://github.com/edwardbair

**Data availability**

Automated in situ measurements are available at: https://snow.ucsb.edu

Sentinel-2A/B MSI imagery is the Copernicus Open Access Hub: https://scihub.copernicus.eu/

Processed in situ measurements and spectroscopic measurements will be placed in a repository such as Zenodo if the article is accepted.

**Author contribution**

According to CRediT taxonomy:

EHB - Conceptualization, data curation, formal analysis, funding acquisition, investigation, methodology, writing (original
draft)

JD - Conceptualization, software, formal analysis, investigation, methodology, writing (review & editing)

CS - Conceptualization, data curation

AL - Resources, funding acquisition

KR - Funding acquisition, writing (review & editing)

AS - Conceptualization, writing (review & editing)

TS - Investigation, writing (review & editing)

RED - Resources, funding acquisition

**Competing interests**

The authors declare that they have no conflict of interest.

**Acknowledgements**

This research was supported by NASA awards: 80NSSC21K0997, 80NSSC20K1722, 80NSSC20K1349, & 80NSSC18K1489. Other support is from Broad Agency Announcement Program and the Cold Regions Research and Engineering Laboratory (ERDC-CRREL) under Contract No.W913E520C0019 and the Department of Defense (DOD) Research Participation Program administered by the Oak Ridge Institute for Science and Education (ORISE).




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
