# Peer review of "Divergence of apparent and intrinsic snow albedo over a season at a sub-alpine site with implications for remote sensing"

_The Cryosphere, 2021_

## Author Comment (AC1)

**Authors' response to RC-1**

The reviewer's comments are in Cambria font.

> The authors' responses are in blue Calibri font and are indented.

The largest issue I have is concerning the description of the methodology, specifically the way the surface roughness is defined and incorporated into the apparent albedo correction. As written, it is hard to determine how the surface roughness data derived from the lidar data were incorporated into the modeled apparent albedo—with the largest confusion occurring over the description of the use of "a generic ablation hollow" to define the surface roughness within the calculation of the rough surface albedo on line 164 in the text vs. the derived distribution of slopes derived from the lidar data used to define a parameter of surface roughness (line 154). I have added specific notes below where this becomes confusing or unclear.

> Instead of point in "a generic ablation hollow," we should clarify that we mean any point on the surface. We treat the rough surface in a similar way that we would consider a digital elevation model (DEM) over mountainous terrain, except in our case, the lidar gives us a DEM at ~1 cm grid spacing. For each point, we calculate the gradient (slope and aspect) along with the angles to the local horizon around the 360° range of azimuths (because some points will be shaded by nearby points). From these values, we calculate the local illumination angle and the sky view factor, the fraction of the overlying hemisphere open to the sky (Dozier, 2022 shows the equations). Unfortunately, we used the term "roughness" to describe generically the rough topography, and then we also defined a specific "surface roughness" (lines 154 and 273 and then lines 279-280 in the caption to Figure 7). We should eliminate this specific surface roughness from Figure 7, which we defined as the "root mean squared value of the slope" because in fact the slopes and horizons AND the solar illumination angle affect the reflected radiation. We apologize for the ambiguous use of the term.

The next issue I have is with the authors' definition of "apparent" and "intrinsic" albedo which is I believe is non-standard within the literature, i.e., apparent albedo typically describes other effects outside of surface roughness, such as slope, aspect and snowpack thickness and geometry, not just surface roughness, and intrinsic albedo typically refers to albedo due to material properties of the snowpack (i.e., effects due to snow grain size, impurities, etc.) and not just a narrow definition of "smooth surfaces." References to the use of apparent and intrinsic albedos referring to rough vs. smooth albedo usage should be included if this is a more common use of the terminology than I am aware of. The rest of the specific comments as follows are minor/technical.

> As described above, by "roughness" we mean the slope, aspect, view factor and the "apparent" albedo as caused by all these attributes of roughness. Snowpack thickness would affect the intrinsic albedo. We need to clarify the definitions (lines 14-15 and 42-44).

Line 54: A very minor suggestion, but the phrase "These models provide intrinsic albedos with lighting conditions controlled by snow properties and illumination angles" is confusing/imprecise as written, as lighting conditions aren't controlled by snow properties.

> Agreed, we will revise to something like "These models provide intrinsic albedos based on grain-scale snow properties, which have included grain shape . . . snow structure . . . effects of light-absorbing particles . . . and vertical heterogeneity . . ."

Line 75 also line 395: This is not a very good description of sastrugi or classification of snow surface roughness features into 3 broad examples. My biggest issue with this description is that it is not generally true sastrugi are smaller than the ablation hollows, at least in polar regions; sastrugi can be on the order of 1m-1.5m in height. Additionally, sastrugi are one form of snow bedforms that also include snow dunes, ripples, scour marks and pits, etc.—and so it might be more correct to include snow bedforms or dunes as a third type of surface roughness feature. I would recommend Filhol and Sturm (2015) as a good review of snow surface roughness features to cite.
Sticking to surface features commonly encountered in mid-latitudes vs. polar areas would avoid needing to delve into some of the complexities when describing sastrugi vs. snow dunes and other polar surface features.

> We appreciate pointing us to the Filhol and Sturm (2015) paper and will revise these paragraphs accordingly.

Line 83: Not sure if the authors are aware of the reference Wright et al., 2014, that discussed apparent albedo effects when making field spectrometer measurements in Greenland in comparison to MODIS retrievals.

> Interesting and relevant paper (Wright et al., 2014). We will incorporate into the discussion here.

Line 95: Would be good to specify how many up and down pointing radiometers of what type there are here to make it less confusing? And to add what sensors are on the fixed arm and what sensors are on the adjustable arm.

> Okay will do. Uplooking and downlooking PSP radiometers with both clear (285-2800 nm) and near-infrared (700-2800 nm) domes are located on both the fixed and adjustable arms, providing redundant measurements of the incoming irradiance in both wavelength regions, and providing measurements of reflected radiation from both the fixed and adjustable arms. The adjustable arm keeps its downlooking radiometers about 1 m above the snow surface, whereas the fixed arm's distance from the snow surface depends on the snow depth. The SP1N Sunshine Pyranometer (400-2700 nm) is mounted only on the fixed arm, and we use those data to estimate the direct and diffuse fractions of the solar irradiance.

Figure 2: Is there an explanation of the offset from the PSP and SPN1 and divergence from modeled results at higher wavelengths for the PSP? Is that a known source of error at higher wavelengths for the PSP?

> Perceptive comment that we need to think about. The difference could be related to the precipitable water vapor in the solar radiation model, as we used the default mid-latitude winter value from SMARTS (Gueymard, 2019) for all dates whereas we could estimate the daily variability. Therefore, we don't truly know whether the "error" is in the SPN1 or PSP measurements.
> (There was some debate among the authors about the necessity of this figure.)

Line 123: For consistency, should add that this "on a computer-controlled and self-leveling arm" is the adjustable arm.

> Agree. Will do.

Figure 3 legend: This is a very minor comment, but I suggest keeping the naming convention used in the legend consistent with Figure 2 and the text.

> Okay. In Figure 2 we will change the legend to put the adjective in front of the radiometer title, i.e. "clear PSP" instead of "PSP clear."

Line 124: This section is confusing as written. If the adjustable arm is kept at 1m, would it be better to specify the heights of the different sensors as 8m and 1m vs. fixed and adjustable? Isn't that the most germane comparison of the measurements that are made with respect to the field of view of the sensors?

> Agree. We will change this sentence to something like, "Reflected radiation was measured with downlooking PSPs, in both broadband (285-2800 nm) and near-infrared (700-2800 nm) wavelengths, mounted on the adjustable arm kept ~1 m above the snow surface and the fixed arm 8 m above the ground, hence its distance above the snow depended on the snow depth. The field-of-view from the fixed arm includes objects other than snow, so using those values to measure snow albedo results in errors that depend on the snow depth.

Line 154: How is this value, i.e., "the root mean squared value of the distribution of slopes in the field-of-view" used in the model in the corrections? Also, suggest adding that this parameter is in degrees to help clarify why degrees are used in Figure 7.

> The RMS value of the slopes is not used at all in the model, and we will eliminate it from the paper.

Line 162: How is a "generic" ablation hollow defined? From the distribution of the slopes derived from lidar point data as described above in line 154? Or an arbitrary ablation hollow based on other past work? Along the same lines, how are the effects averaged over the footprint, i.e., how was the spacing between ablation hollows determined—from Lidar data or some other means? There would be a sensitivity due to spacing, size, etc. Also, how is the temporal nature of the surface roughness treated since these features will change over time?

> Poor phrasing on our part. A "generic point" on the rough surface has an elevation. From the elevations of its immediate neighbors we derive its gradient, and from the elevations of all other points we determine the angles to the point's horizon in all directions. The lidar images the area every hour, so indeed the topography changes, and we calculate the topographic parameters daily.

Line 173: typo/missing word somewhere: the initial approach the model this effect uses monochromatic radiation, "is" in front of monochromatic?

> Thanks, will change to "the initial approach **to** model this effect . . ."

Line 175: Define variable I upfront in description, it is slightly confusing here as written if I is equal to irradiance or initial irradiance (suggest rewording to the initial irradiance I is set to 1 perhaps) gets a little sloppy/messy with the definitions for irradiance and hard to follow, i.e., line 209 I is defined as spectral radiation, and then I$_{reflected}$ is used in Equation 8

Ah, okay. We can eliminate the $I = 1$ in lines 175 and 192 and incorporate $I$ in equation (4). Then the rest of the equations follow.

Line 197: what is the term/variable "alphaI" referring to?

Sorry, should write it as $\alpha \times I$ to be clearer.

Line 205: Why was San Juan dust mass concentration used in the model and is that valid for this region? Was there mass concentration or other info collected in the Sterle et al., 2013 study mentioned in line 247?

We use the optical properties of the San Juan dust and solve for the mass concentration for each day.

Line 233: missing word (maybe "model") after SPIReS definition as it is awkward as written

We will insert "model" after Sensing. Thanks.

Line 236: For future work, possible to do lidar surveys over the target pixel? Seems like the lidar data are under utilized for the remote sensing validation portion for this work. Is the slope and/or aspect at least similar to the CUES pixel?

Probably not. The lidar is not really portable, and it is mounted on a tower (Figure 1) that is outside the area open to skiers. The slope of the target pixel is nearly flat, as is the CUES pixel. The ground beneath the snow in the CUES pixel is indeed flat, but during the season the wind causes the snow to drift, so the slope and aspect varies.

Line 260: I think there is a mistake in the text as this seems to describe Figure 6, not Figure 5.

The reviewer's comment is correct.

Line 270: should be "is also shown" since the subject is "an unadjusted fsca"

The reviewer's comment is correct.

Line 271: Is the surface roughness the distribution calculated from the lidar data or is the "generic ablation hollow"? (see comment from section 2.2 about this)

We will use "surface roughness" or "roughness" itself as a generic description of the snow surface. We eliminate reference to a calculated "surface roughness" because the model does not used the value so defined ("root mean squared value of the slope of the snow surface").

Figure 7: Suggest using two separate scales for albedos and for the surface since 30 is an awkward divider at best for the reader to interpret, and the units are different the way the caption is written since the caption is implying the surface roughness is still in degrees after dividing by a scalar 30. Please ignore suggestion if it makes the plot too busy, but refine the caption to reflect the units. Also note the suggestion up at line 154 to include a phrase noting that the distribution of the slopes derived from the lidar data are in degrees in order to make it clearer what value is plotted here (see comment about line 271---both the surface roughness derived from the lidar data and the "generic ablation hollow" concept are described in section 2.2).

As noted just above, we will eliminate the surface roughness from this figure.

Line 276: In the caption seems like there is a typo here, "with the error bars (2.0 %) representing based on stated values from the manufacturer." And it should be "with the error bars (2.0 %) representing stated values from the manufacturer." Or "with the error bars (2.0 %) based on stated values from the manufacturer."

Agree. We will use "with the error bars (2.0 %) based on stated values from the manufacturer."

Line 395: As mentioned before, sastrugi can reach heights of 1-1.5m in polar areas and so this statement is not correct.

Agree. See our earlier comment, "We appreciate pointing us to the Filhol and Sturm (2015) paper and will revise these paragraphs accordingly.

References cited in the Response:

Dozier, J.: Revisiting the topographic horizon problem in the era of big data and parallel computing, IEEE Geoscience and Remote Sensing Letters, 19, 8024605, https://doi.org/10.1109/LGRS.2021.3125278, 2022.

Filhol, S., and Sturm, M.: Snow bedforms: A review, new data, and a formation model, Journal of Geophysical Research: Earth Surface, 120, 1645-1669, https://doi.org/10.1002/2015JF003529, 2015.

Gueymard, C. A.: The SMARTS spectral irradiance model after 25 years: New developments and validation of reference spectra, Solar Energy, 187, 233-253, https://doi.org/10.1016/j.solener.2019.05.048, 2019.

Wright, P., Bergin, M., Dibb, J., Lefer, B., Domine, F., Carman, T., Carmagnola, C., Dumont, M., Courville, Z., Schaaf, C., and Wang, Z.: Comparing MODIS daily snow albedo to spectral albedo field measurements in Central Greenland, Remote Sensing of Environment, 140, 118-129, https://doi.org/10.1016/j.rse.2013.08.044, 2014.

---

## Author Comment (AC2)

**Authors' Response to RC-2**

The reviewer's comments are in Cambria font.

> The authors' responses are in blue Calibri font and are indented.

Line 151: How do you compute slope and aspect based on a radial mask? Could you give an example ideally with some illustrations? Do you "divide" the snow surface into numerous hollows that the "slope" is the slope of each hollow?

> We treat the rough surface in a similar way that we would consider a digital elevation model (DEM) over mountainous terrain, except in our case, the lidar gives us a DEM at ~1 cm grid spacing. For each point, we calculate the gradient (slope and aspect) along with the angles to the local horizon around the 360° range of azimuths (because some points will be shaded by nearby points). From these values, we calculate the local illumination angle and the sky view factor, the fraction of the overlying hemisphere open to the sky (Dozier, 2022 shows the equations). The "radial mask" is the footprint of the downlooking radiometer; the slopes and aspects vary with the topography **within** the radial mask.

Line 154: It seems surface roughness only accounts for the distribution of slopes? What about the aspect? Given a mask, could you provide an example of how to count the distribution and compute surface roughness?

> Unfortunately, we used the term "roughness" to describe generically the rough topography, and then we also defined a specific "surface roughness" in line 154. We will eliminate that sentence and also fix line 273 and lines 279-280 in the caption to Figure 7.

Line 159: Are downwelling direct irradiance B and diffuse irradiance D used here measured or modeled?

> Their ratios to total $I_\downarrow$ are measured by the SP1N Sunshine Pyranometer and then applied to the PSPs (which have a slightly different wavelength range).

Line 162: What is a "generic ablation hollow", how is this defined in this work?

> Instead of "a generic ablation hollow," we should clarify that we mean any generic point on the surface, which has an elevation, a gradient (slope and aspect), and a sky view factor.

Line 164 - 166: Similar to the illustration of surface roughness, it is valuable and helpful to provide a figure on illumination angle.

> We can, although this equation for illumination angle is standard. We think if we specify something like "adjoining elevations," we can eliminate the confusion we caused by our phrase "generic ablation hollow."

Line 198-202: The description of the method is confusing here. For example, what is the modeled average $\overline{\alpha_{spatial}}$?

> $\alpha_{spatial}$ is modeled at every point in the 1 cm topographic grid. $\overline{\alpha_{spatial}}$ is the mean of those values over the field-of-view of the downlooking radiometer, and it compares with the measured albedo from that radiometer.

Line 204: The authors describe the spectral albedo simulations here, while the "simulated albedo" has already been mentioned multiple times in the previous text. Consider rearranging the text so readers understand what is simulated albedo before its being used.

> We should use "modeled" rather than "simulated."

Line 205-206: Why assume San Juan dust with an effective radius of 3 microns?

> Among the values for the complex refractive index of dust in the SNICAR model, the San Juan dust is the one in the western U.S. The 3 µm radius corresponds to samples we collected and measured by a colleague at the Desert Research Institute. We are working on a separate paper involving those measurements.

Figure 4: This is an interesting Figure that requires some details. Mainly, what caused the spread of initial albedo in the x-axis? Snow depth? Grain size? Impurities? Spectral distribution of downwelling flux? What is the roughness of this case?

> The spread in the *x*-axis covers the range of values of the intrinsic albedo we encountered during the experiment, as governed by the grain size and concentration of the light-absorbing particles.

Please also discuss why the albedo increase is more significant when initial albedos are roughly within 0.68-0.80.

> We probably have to look at the details of each step in the re-reflection to address this. We speculate that this range of albedo values has a large range of spectral albedos that cause it. With each reflection, the lower spectral albedos are absorbed.

Section 2.1: how often did one adjust the adjustable arm for measurements? Was the goal of each adjustment to maximum the snow coverage in the field of view?

> In measuring the reflected radiation, two artifacts must be minimized. If the downlooking radiometer is too far above the snow, the field-of-view is too large (nearly 9×*h*, where *h* is in meters) so other, darker elements like the tower itself and trees, will cause the albedo to be too low. Conversely, if the radiometer and its arm are too close to the snow, they will cast a shadow that will also cause the albedo to be too low. By experiment, we found that the combination of these two artifacts is minimized when the radiometer is ~1 m above the snow, so as the snow depth changes, we maintain the radiometers' height.

Line 281: It seems Figure 6 is discussed after Figure 7; please consider swapping the Figure label.

> Our error. The reference to Figure 5 on line 260 should be to Figure 6.

**Reference cited in the Response**

Dozier, J.: Revisiting the topographic horizon problem in the era of big data and parallel computing, IEEE Geoscience and Remote Sensing Letters, 19, 8024605, https://doi.org/10.1109/LGRS.2021.3125278, 2022.